# The Right Approach: Power of Biomarkers in the Assessment and Management of Right Ventricular Dysfunction

**DOI:** 10.3390/ijms26189064

**Published:** 2025-09-17

**Authors:** Mihajlo Viduljević, Marija Polovina, Oliviana Geavlete, Marianna Adamo, Adi Hadžibegović, Milika Ašanin, Sanja Stanković, Tuvia Ben Gal, Mohamed A. Abdelwahab, Magdy Abdelhamid, Andrew P. Ambrosy, Ovidiu Chioncel, Petar M. Seferović

**Affiliations:** 1Emergency Medicine and Center for Medical Biochemistry, Department of Cardiology, University Clinical Centre of Serbia, 11000 Belgrade, Serbia; mihajloviduljevic@gmail.com (M.V.); a.hadzibegovic@gmail.com (A.H.); masanin2013@gmail.com (M.A.); sanjast2013@gmail.com (S.S.); 2Faculty of Medicine, University of Belgrade, 11000 Belgrade, Serbia; seferovic.petar@gmail.com; 3Emergency Institute for Cardiovascular Diseases Prof. C.C. Iliescu, University of Medicine Carol Davila, 022328 Bucharest, Romania; oliviana_m@yahoo.com (O.G.); ochioncel@yahoo.co.uk (O.C.); 4Institute of Cardiology, Department of Medical and Surgical Specialties, Radiological Sciences and Public Health, University of Brescia, 25123 Brescia, Italy; mariannaadamo@hotmail.com; 5Department of Biochemistry, Faculty of Medical Sciences, University of Kragujevac, 34000 Kragujevac, Serbia; 6Heart Failure Unit, Cardiology Department, Rabin Medical Center, Faculty of Medicine, Tel Aviv University, Tel Aviv 4941492, Israel; bengaltu@gmail.com; 7Department of Cardiovascular Medicine, Faculty of Medicine Kasr Al-Ainy, Cairo University, Cairo 4240310, Egypt; mohamed.a.adel@kasralainy.edu.eg (M.A.A.); magdyabdelhamid@hotmail.com (M.A.); 8Division of Research, Kaiser Permanente Northern California, California, CA 94588, USA; andrew.p.ambrosy@kp.org; 9Serbian Academy of Sciences and Arts, 11000 Belgrade, Serbia

**Keywords:** biomarker, cardiovascular disease, right ventricle, dysfunction, heart failure, acute myocardial infarction, pulmonary embolism, pulmonary hypertension, diagnosis, risk stratification, therapy

## Abstract

Right ventricular (RV) dysfunction is common and linked to poor outcomes across conditions such as heart failure (HF), acute coronary syndromes, pulmonary embolism, and pulmonary hypertension. While imaging, electrocardiogram (ECG), and invasive tests remain central to RV assessment, circulating biomarkers offer a rapid, non-invasive, and reliable alternative. These biomarkers reflect key pathophysiological processes, including myocardial injury, stress, fibrosis, inflammation, congestion, and multiorgan involvement. High-sensitivity troponins and natriuretic peptides (BNP, NT-proBNP) are already widely used, while emerging biomarkers—such as CA125, copeptin, galectin-3, and others—may enhance diagnostic accuracy and risk stratification. Some, like CA125 and NT-proBNP, have shown promise in guiding post-discharge therapy. However, challenges remain regarding the specificity of biomarkers for RV dysfunction and their role across different clinical contexts. This review provides an integrated overview of RV dysfunction, with a focus on the diagnostic and therapeutic potential of both established and novel biomarkers.

## 1. Introduction

Right ventricular (RV) dysfunction is common across different clinical settings and associated with the more severe clinical presentation, greater arrhythmic propensity, worse functional status, and higher risk of hospitalisation for heart failure (HF) and mortality [1,2,3]. The presence of RV dysfunction may hamper the tolerance of guideline-directed medical therapies (GDMT) for HF, and thus impact the clinical outcomes [4]. Despite its clinical and prognostic significance, RV dysfunction frequently remains overlooked, and its timely diagnosis remains a challenge. Current diagnostic strategies principally rely on imaging and invasive assessments [5]. Laboratory biomarkers offer rapid, reliable, and non-invasive tools that can be used in diagnostic assessment, risk stratification, prognostication and management of RV failure. Cardiac troponins and natriuretic peptides remain the most widely used biomarkers. However, growing interest is being directed towards novel biomarkers that capture diverse aspects of RV pathophysiology and its systemic consequences (e.g., carbohydrate antigen 125 (CA125) heart type fatty acid–binding protein (hFABP), growth differentiation factor 15 (GDF15), galectin-3 etc.) [6]. The present review aims to address the key aspects of the pathophysiology, clinical presentation, diagnosis, and management of RV dysfunction, with a focus on the role of the established and emerging biomarkers in the evaluation and treatment of RV dysfunction in different clinical situations.

## 2. Epidemiology, Pathophysiology and Prognosis of Right Ventricular Dysfunction

According to the European Society of Cardiology Heart Failure Long Term Registry, RV dysfunction is present in 3.5% of acute HF admissions and is associated with adverse prognosis, with in-hospital mortality rate of 9.4% [7]. Notably, one-year all-cause mortality and hospitalizations rates in patients with RV dysfunction were 33.9% and 48.3%, respectively [7]. Mortality in patients with acute myocardial infarction complicated by RV involvement is higher than in patients without RV involvement and increases as the condition deteriorates into cardiogenic shock [8]. In acute pulmonary embolism, RV dysfunction can be detected in 40–70% of patients, and its presence confers up to two fold-higher mortality [9]. In chronic HF, the prevalence of RV dysfunction is estimated at 48% of patients with HF and reduced ejection fraction (HFrEF), and 20–40% of those with HF and preserved ejection fraction (HFpEF) [2,10]. Even in patients without known HF, evidence of reduced RV contractility is associated with increased risk of incident HF or death independently of other factors [11].

The most significant underlying aetiologies and mechanisms of RV dysfunction are summarised in Figure 1.

The pathophysiology of RV—dysfunction encompasses three main direct mechanisms, which frequently overlap [12], and include (1) increased RV afterload, (2) reduced RV contractility and/or (3) increased RV preload, Figure 1.

### 2.1. Increased Afterload

The RV is sensitive to an increase in afterload due to its relationship with the pulmonary vascular network, which is normally characterised by low vascular resistance. In the acute setting (i.e., acute HF, pulmonary embolism, pneumonia, acute respiratory distress), the RV responds by dilating and increasing its contractility, but RV failure may occur rapidly if these mechanisms prove insufficient [5]. In the chronic setting (i.e., chronic left-sided HF, pulmonary hypertension, chronic pulmonary disorders), the RV undergoes progressive hypertrophy, enabling it to preserve cardiac output. However, it eventually dilates, resulting in tricuspid regurgitation and a subsequent decline in cardiac output [5]. This adaptive response is particularly relevant in chronic states associated with pulmonary hypertension, where the RV-pulmonary artery coupling is essential for maintaining cardiac output [13].

### 2.2. Reduced Contractility

Acute reduction in RV contractility is mostly caused by acute myocardial ischemia/infarction or acute inflammation (i.e., myocarditis) [14,15]. In sepsis and septic shock, the systemic inflammatory response was associated with a high prevalence (48%) of RV dysfunction, defined by echocardiographically assessed fractional area change < 35% or tricuspid annular plane systolic excursion < 16 mm [16]. RV dysfunction is associated with a more complicated clinical course and reduced 30-day survival [16,17]. Moreover, RV involvement has been reported in patients with left-sided takotsubo syndrome, with isolated RV forms also documented, typically presenting with more severe clinical manifestations [18,19]. Chronic RV dysfunction most commonly accompanies left-sided HF but may also result from chronic pressure or volume overload or intrinsic myocardial diseases such as cardiomyopathies or cardiac sarcoidosis [20]. In addition to reduced cardiac output, dilated and hypocontractile RV contributes to left ventricular (LV) dysfunction through ventricular interdependence. The overloaded RV “competes” with the LV for the space within the pericardium, leading to the displacement of the interventricular septum towards the LV, impairment in LV filling, and further reduction in cardiac output. Furthermore, elevated pressures in RV free wall, and a decrease in systemic blood pressure, can impact the coronary blood flow, and precipitate worsening RV failure [12].

### 2.3. Increased Preload

The RV is highly dependent on the appropriate preload for its optimal function. However, excessive RV preload due to renal and/or hepatic failure, severe tricuspid regurgitation, congenital heart diseases (i.e., atrial or ventricular septal defect, partial or total anomalous pulmonary venous return, Ebstein anomaly, patent ductus arteriosus) or iatrogenic volume overload can lead to deterioration in RV function [12]. Significant bradycardia can also contribute to an increase in RV preload. In addition, increased LV output and venous return following implantation of a left ventricular assist device (LVAD) acutely increase RV preload and can precipitate RV failure.

Of note, irrespective of the primary pathophysiological mechanism of RV dysfunction, secondary tricuspid regurgitation, resulting from RV dilatation and dysfunction, amplifies both the hemodynamic disturbances and the clinical deterioration, fuelling a vicious cycle that leads to poorer patient outcomes [21].

## 3. Diagnostic Assessment of Right Ventricular Dysfunction

The clinical suspicion of RV dysfunction should be raised in the setting of acute and chronic conditions frequently complicated with RV failure (Figure 1). Likewise, in all cases of chronic HFrEF or HFpEF, a careful search for the signs and symptoms of RV failure should be performed. The most important clinical signs and symptoms and electrocardiographic features of RF dysfunction are presented in Table 1.

Imaging modalities are essential for the evaluation of RV dysfunction [5,12,23]. The diagnostic process typically begins with focus bedside echocardiography, followed, if necessary, by a more comprehensive assessment. The most important echocardiographic characteristics of RV dysfunction are presented in Table 2.

Further assessment with cardiac and pulmonary artery computed tomography (CT) and cardiac magnetic resonance (CMR) imaging play a valuable role in the evaluation of specific aetiologies and characterisation of intramyocardial processes in acute and chronic settings (e.g., pulmonary embolism, valvular heart disease, cardiomyopathies, myocarditis, sarcoidosis etc) [15,22,24]. In selected patients, right heart catheterisation with invasive haemodynamic assessment may be considered [25,26]. Endomyocardial biopsy should be considered in patients presenting with suspected fulminant myocarditis, severe acute HF unresponsive to conventional management, and in clinical situations where the aetiology of HF remains unresolved following non-invasive evaluation [27].

## 4. Biomarkers in Diagnostic and Prognostic Evaluation of Right Ventricular Disfunction

The diagnostic and prognostic role of biomarkers in RV dysfunction is increasingly acknowledged, driven by the emergence of circulating markers that reflect underlying pathophysiological processes and their systemic impact. Some of the established and emerging biomarkers validated for the assessment of RV dysfunction are presented in Figure 2.

### 4.1. Biomarkers of Myocardial Injury

Cardiac troponins I and T are the most validated and utilized biomarkers of myocardial injury (Figure 2). Current recommendation is to assess these biomarkers with the use of high-sensitivity assays, which can detect very small serum troponin concentrations, very early during myocardial injury [28]. Diagnostic utility of cardiac troponins has been best characterized for patients with suspected acute myocardial infarction. Since high-sensitivity assays can detect a rise in cardiac troponins within one to two hours of myocardial injury, a rule-in/rule-out algorithm, integrating clinical assessment and early troponin testing, has been implemented to expedite the diagnosis of suspected acute coronary syndrome [29]. Both high-sensitivity cardiac troponin I and T can be used, as their early serum concentrations correlate linearly [30]. There are currently no data specifically indicating the prognostic role of cardiac troponins in RV myocardial infarction, but higher admission levels of cardiac troponin troponins predicted a greater risk of subsequent cardiac events and mortality in patients with acute myocardial infarction [31].

Elevated cardiac troponins also play a significant role in assessing the patients with acute pulmonary embolism. Firstly, they provide independent, additional diagnostic value for identifying RV dysfunction, complementing findings from echocardiography and pulmonary artery CT [32,33]. Furthermore, a meta-analysis demonstrated that elevated levels of cardiac troponins (both troponin I and T), are associated with significantly increased in-hospital mortality, even among normotensive individuals [34]. A concentration-dependent relationship between cardiac troponins and adverse long-term outcomes has been established following pulmonary embolism [35]. However, when used on their own, cardiac troponins have relatively low specificity and positive predictive value for early mortality, particularly in normotensive patients [36] Their detection, therefore, has been integrated with clinical and imaging assessment, into a comprehensive grading of disease severity, which is particularly relevant for the management and prognostication of apparently haemodynamically stable, normotensive patients [36]. Accordingly, normotensive patients who exhibit clinical signs of disease severity and have elevated cardiac troponin levels should be considered at intermediate risk, even if echocardiographic or CT evidence of RV dysfunction is absent [36]. Those normotensive patients with pulmonary embolism, who have clinical signs of the more severe disease (e.g., increased pulmonary embolism severity index score) along with elevated troponin and imaging evidence of RV dysfunction, should be regarded as intermediate-high risk [36]. On the other hand, high-sensitivity troponin level below the upper limit of normal, has a high negative predictive value for adverse outcomes [37]. Age-adjusted troponin T cut-off values (≥14 pg/mL for patients aged <75 years and ≥45 pg/mL for those ≥75 years) may further improve the negative predictive value of this biomarker [38].

Elevated levels of cardiac troponins can be detected in non-ischaemic causes of acute and chronic RV dysfunction, including acute myocarditis, right-sided or biventricular takotsubo syndrome, sepsis, acute respiratory insufficiency/distress, as well as in chronic HF, cardiomyopathies and/or pulmonary hypertension [39]. In patients with pulmonary hypertension, elevated troponin levels predict poor outcomes regardless of the clinical type of this condition [40]. In patients with arrhythmogenic RV cardiomyopathy (ARVC), the clinical course is often marked by “hot phases”, characterised by chest pain and elevated levels of troponin and inflammatory biomarkers in the absence of obstructive coronary artery disease. Endomyocardial biopsy specimens reveal myocardial inflammation in a significant proportion of patients experiencing “hot phases”, possibly resulting from the instability of junctional proteins due to pertinent genetic mutations, which trigger inflammatory response with a release of proinflammatory mediators (tumour necrosis factor alpha and interleukin-6) and troponin, reflecting myocardial injury [41].

Elevated levels of cardiac troponin can be observed in patients with extracardiac conditions, in particular chronic kidney disease and type 2 diabetes, which frequently coexist with HF and RV dysfunction. These conditions contribute to chronic myocardial injury, characterised by persistent and stable elevation of cardiac troponin levels above the upper limit of normal [42]. Troponin concentrations (especially troponin T) can be further raised by reduced renal clearance. Of note, chronic myocardial injury has been associated with significantly impaired long-term prognosis, regardless of the cause [43], and across all levels of kidney function [44].

In addition to cardiac troponin, h-FABP has emerged as a promising biomarker in the evaluation of suspected RV dysfunction (Figure 2). h-FABP is a cytoplasmic protein expressed in tissues with high lipid metabolism, such as the myocardium. Due to its small molecular size, it is rapidly released into the bloodstream (typically less than one hour) following myocardial injury, reaching peak levels at 6 to 8 h and normalising within 24 to 36 h [45]. Being an early and sensitive biomarker of myocardial injury, it has been mostly assessed in the setting of RV dysfunction due to acute pulmonary embolism. It provides additive prognostic information to standard care, both in unselected and normotensive patients with acute pulmonary embolism [46,47]. A meta-analysis identified h-FABP concentrations ≥ 6 ng/mL to be associated with adverse short-term outcomes and all-cause death in acute pulmonary embolism [48]. Furthermore, h-FABP was assessed among patients with chronic HF involving RV dysfunction. A cohort study suggested that h-FABP could have an incremental predictive value to natriuretic peptides in detecting individuals with chronic HF at high risk of adverse events [49]. A poor correlation between the two biomarkers (i.e., h-FABP and natriuretic peptides) may be caused by different release mechanisms, strengthening utility of their combined use [49].

### 4.2. Biomarkers of Myocardial Stress

Pressure and/or volume overload of the RV is associated with myocardial stretch, which causes the release of B-type natriuretic peptide (BNP) and N-terminal (NT)-proBNP (Figure 2). In the setting of acute RV dysfunction in pulmonary embolism, elevated BNP or NT-pro-BNP levels indicated a higher risk of in-hospital complications and 30-day mortality, including normotensive patients [50,51]. As with troponin, it is recommended to integrate clinical presentation, imaging evidence of RV dysfunction and testing of multiple biomarkers in the risk assessment [36].

In patients with acute HF, BNP and NT-proBNP have a validated role for early diagnosis or exclusion of acute HF, particularly in cases of clinical uncertainty. For suspected acute HF, cut-off concentrations of BNP and NT-proBNP of <100 pg/mL and <300 pg/mL, respectively, have demonstrated high negative predicative value for the rule out [52]. However, several clinical situations involving RV dysfunction can be associated with lower-than-expected levels of natriuretic peptides. Firstly, in patients with isolated RV dysfunction, natriuretic peptide levels may appear lower relative to the degree of systemic congestion, due to the comparatively smaller myocardial mass of the RV [53]. Likewise, in acute decompensated HFpEF, natriuretic peptide concentrations may be lower than anticipated given the extent of volume overload [53]. A similar situation may occur in obese individuals with RV dysfunction who tend to exhibit lower levels of natriuretic peptides, attributed to a complex interplay of reduced cardiac synthesis and enhanced clearance by adipose tissue [54,55]. These situations may result in misinterpretation and a delay in diagnosis and treatment.

In the chronic setting, natriuretic peptides also play a key role in the evaluation of patients with suspected HF [52]. Current guidelines recommend a diagnostic cut-off value of <35 pg/mL for BNP and <125 pg/mL for NT-proBNP in the ambulatory setting for ruling out HF regardless of age, or its clinical type [52]. Like acute HF, in patients with chronic HFpEF, or those with isolated RV dysfunction, these values may be lower than in HFrEF [56]. A recent analysis suggested that the recommended rule-in/rule-out NT-proBNP threshold < 125 pg/mL may have low sensitivity and specificity in suspected HFpEF patients, particularly with obesity or atrial fibrillation [57]. It was suggested that the application of a separate rule-in and rule-out diagnostic thresholds stratified by body mass index may improve diagnostic precision [57]. The additive diagnostic value of natriuretic peptides in atrial fibrillation are less clear because of its frequent co-occurrence with HFpEF and RV dysfunction [57]. The role of natriuretic peptides has been established for the evaluation of chronic RV dysfunction in pulmonary hypertension, where a diagnostic algorithm integrating symptoms, carbon monoxide diffusion capacity and NT-proBNP identified subjects at very low probability of pulmonary hypertension, who may be deferred from further evaluation [58]. However, although natriuretic peptides reflect increased RV wall stress, their specificity for RV dysfunction is limited, particularly in the context of biventricular failure. Interpretation may also be confounded by factors such as advanced age, renal impairment, and other comorbidities, all of which should be carefully considered for patient evaluation [59].

Copeptin also emerged as a promising biomarker for assessing RV dysfunction (Figure 2). Copeptin is the C-terminal part of vasopressin prohormone and is released in stressful situations, including myocardial stress. Its levels have been found to correlate with echocardiographic indicators of reduced RV longitudinal function [60]. Concentrations ≥ 24 pmol/L have been associated with echocardiographically confirmed RV dysfunction and may aid in evaluating its severity [61].

### 4.3. Biomarkers of Myocardial Remodelling and Fibrosis

The development of HF is frequently driven by molecular pathways that promote myocardial remodelling and fibrosis, particularly in cardiomyopathies, pulmonary hypertension, systemic sclerosis and HFpEF [62]. Therefore, biomarkers that reflect fibrotic processes may offer valuable insights into disease assessment and progression of HF. Among the most extensively studied biomarkers reflecting fibrosis are soluble suppression of tumorigenicity-2 (sST2), GDF-15, galectin-3, and circulating microRNAs (Figure 2). sST2 is a soluble isoform of the interleukin-1 receptor. It binds interleukin-33, antagonizing its protective, anti-fibrotic effects on the myocardium, thus promoting hypertrophy and fibrosis [63]. Current evidence suggests that sST2 represents an independent predictor of mortality that can modestly improve risk stratification models in patients with HF and RV dysfunction and is less affected by renal function than natriuretic peptides [64,65]. GDF-15 is a cytokine of the transforming growth factor-β family, upregulated in response to oxidative stress, myocardial injury, and inflammation. In patients with advanced HFrEF, higher levels of GDF-15 were associated with more severe RV dysfunction and worse congestion, cachexia, and a higher risk of adverse events independently of other characteristics [66]. Galectin-3 is secreted by macrophages involved in inflammation, and cardiac remodelling. A cohort study suggested that among patients with HFrEF, those with higher galectin-3 levels had more pronounced RV dysfunction and greater exercise intolerance [67]. Finally, microRNAs represent small non-coding RNAs involved in post-transcriptional gene regulation, which are released into circulation during cardiac stress and remodelling. These biomarkers were associated with RV hypertrophy and fibrosis, particularly in cardiomyopathies and pulmonary hypertension [68].

### 4.4. Biomarkers of Congestion, Systemic Inflammation, and Hypoxia

Congestion is a hallmark of HF, and RV dysfunction is typically associated with systemic congestion of the splanchnic circulation, liver, kidneys, and peripheral oedema. Although frequently perceived as a hemodynamic problem, congestion is increasingly recognised as a systemic condition that impairs multi-organ function and contributes to systemic inflammation [63]. In more severe cases of RV and/or biventricular dysfunction accompanied by hypoperfusion, and cardiogenic shock, organ and tissue hypoxia further exacerbate functional impairment. Therefore, the role of biomarkers of congestion, systemic inflammation and hypoxia has been increasingly recognised in the evaluation of patients with HF and RV dysfunction.

CA125 is a large glycoprotein synthesized by mesothelial cells in the pericardium, pleura, and peritoneum. It has been long recognised as a biomarker of ovarian and other malignancies, but it has recently emerged as a biomarker of congestion and inflammation in HF, particularly involving the RV (Figure 2). Although the underlying mechanisms its role in congestion have not been fully elucidated, it seems that its secretion is stimulated by increased hydrostatic pressures, and exposure of mesenchymal cells to proinflammatory cytokines (interleukin 1, tumour necrosis factor-α, and lipopolysaccharide) [69]. In a study of patients with acute HF, CA125 has shown greater specificity for RV dysfunction compared to NT-proBNP [70]. Indeed, the major correlates of the rise in CA125 were the clinical parameters of congestion and the severity of tricuspid regurgitation (a proxy of RV dysfunction), fortifying its value as a more specific biomarker for right-sided HF [70]. Elevated levels of CA125 can help expedite the diagnosis of acute HF in cases where natriuretic peptide concentrations are lower than expected, such as in isolated RV dysfunction, HFpEF, or obesity [53]. Higher values of CA125 in patients with chronic obstructive pulmonary disease have demonstrated good correlation with RV dysfunction, suggesting this biomarker as a candidate for guiding referral for echocardiographic exam of high-risk individuals [71]. Furthermore, CA125 has shown incremental prognostic value to NT-proBNP in the assessment of long-term outcomes in patients with HF [72]. The poorest prognosis was seen in patients with sustained elevations of both biomarkers, while those with elevation of only one had an intermediate risk [72]. The most favourable outcomes were observed in individuals with low levels of both biomarkers after an episode of decompensated HF [72].

Biologically active adrenomedullin (bio-ADM) has also been investigated as a biomarker of congestion (Figure 2). Bio-ADM is a peptide hormone produced by various tissues, predominantly by endothelial and vascular smooth muscle cells [73]. Its primary functions include mediating vasodilation and preserving the integrity of the vascular endothelial barrier [73]. In a study of patients with acute HF, bio-ADM was the strongest predictor of the clinical congestion score, particularly reflecting signs of systemic venous congestion associated with RV dysfunction [74]. Higher bio-ADM values also predicted poor up-titration of renin-angiotensin system inhibitors and higher risk of HF readmission and all-cause mortality at 3 months [74]. This biomarker was also shown to outperform natriuretic peptides in predicting residual congestion, one of the main reasons for HF readmissions following hospital discharge [75]. Another study employing right heart catheterisation in advanced HF patients, demonstrated a correlation between bio-ADM values and biventricular filling pressures with the strongest correlation with right atrial pressure [76].

Circulating biomarkers of systemic inflammation have also been explored in the context of different HF phenotypes (Figure 2). While biomarkers of myocardial stress and remodelling (e.g., NT-proBNP, GDF-15) are more strongly associated with HFrEF, inflammatory markers such as high-sensitivity C-reactive protein (CRP), beta-2 integrin and catenin beta-1 show a closer link with HFpEF, likely reflecting distinct underlying pathophysiology and the influence of comorbidities driving systemic inflammation in HFpEF [77]. Biomarkers of low-grade inflammation (e.g., high-sensitivity CRP, tumour necrosis factor-alpha, lipoprotein associated phospholipase A2, and adiponectin), were associated with echocardiographic signs of RV dysfunction in patients with type 2 diabetes [78]. High levels of plasma oxidised low density lipoprotein were proposed as novel indicators of myocardial fatty infiltration, RV dysfunction, and major arrhythmic risk in patients with ARVC [79]. The role of other inflammatory biomarkers in assessing RV dysfunction remains limited and warrants further confirmation.

In severe HF and cardiogenic shock, organ hypoperfusion and tissue hypoxia trigger anaerobic metabolism, leading to an elevation in plasma lactate levels (Figure 2). Although mechanisms of lactate rise in this setting extend beyond tissue hypoxia, lactate levels >2 mmol/L have been established as a biomarker of circulatory and cellular/metabolic abnormalities in overt cardiogenic shock [80].

### 4.5. Biomarkers of Renal Impairment

Renal dysfunction is frequently associated with HF, particularly in the setting of RV dysfunction [81]. Even minor variations in cardiac output and congestion can significantly affect renal perfusion by decreasing renal blood flow and increasing renal venous pressure [81]. The resulting activation of neurohormonal, sympathetic, oxidative, and inflammatory pathways contributes to the progression of both renal and cardiac dysfunction, further aggravated by volume overload, anaemia, and albuminuria [81]. Therefore, biomarkers of renal dysfunction have gained significant appreciation in the clinical assessment of patients with HF (Figure 2). In addition to creatinine-estimated glomerular filtration rate (eGFR), which remains the most validated renal biomarker, cystatin-C has been shown to provide a more reliable estimation of renal function, unaffected by age, sex, or body mass [81]. It has also been shown to predict adverse outcomes in patients with HF and may be valuable in the assessment of RV dysfunction [82]. In a study of patients with pulmonary hypertension who underwent extensive evaluation including right heart catheterisation, cystatin C outperformed NT-proBNP in correlating with pulmonary artery pressure and RV morphology and function, independently of renal function [83]. Another study identified cystatin C as an independent predictor of adverse outcomes in patients with precapillary pulmonary hypertension [84]. Cystatin C levels ≥1.0 mg/L along with advanced functional class impairment identified individuals at particularly high-risk of future events [84]. Albuminuria is also highly prevalent in patients with HF and strongly associated with adverse outcomes [85]. Specific data on patients with RV failure are missing, but low-grade albuminuria has been observed in individuals with pulmonary hypertension and evidence of RV dysfunction, and associated with higher risk of adverse outcomes [86]. Finally, neutrophil gelatinase-associated lipocalin (NGAL), considered to represent a biomarker of acute tubular injury, has been shown as an early and reliable predictor of the development of acute kidney injury and cardiorenal syndrome in acute HF (more commonly observed in those with concomitant RV dysfunction), suggesting that it may be useful in early identification of patients at risk [87].

### 4.6. Biomarkers of Liver Dysfunction

In patients with RV dysfunction, liver abnormalities are common and reflect the hemodynamic consequences of chronic congestion and/or hypoperfusion. Biomarkers such as alanine aminotransferase (ALT), aspartate aminotransferase (AST), lactate dehydrogenase (LDH), gamma-glutamyl transferase (GGT) and alkaline phosphatase (ALP) are typically elevated in the context of hepatic congestion or acute hepatocellular injury [88]. The rise in aminotransferases usually occurs 1 to 3 days after hemodynamic deterioration, and an ALT to LDH ratio < 1.5 is suggestive of cardiogenic acute liver injury [89]. Bilirubin levels, particularly direct bilirubin, increase primarily due to cholestasis from venous congestion. Hypoalbuminemia and bleeding diathesis (due to a deficit in coagulation factors) reflect impaired synthetic liver function, often as a consequence of prolonged hypoperfusion or advanced liver injury [90].

Registry data, including findings from RO-AHFS have demonstrated that elevated admission levels of AST and ALT (more frequently observed with RV dysfunction, lower ejection fraction or cardiogenic shock) are associated with a higher risk of complications during hospitalisation for acute HF [91]. Moreover, marked elevation of ALT is independently associated with increased in-hospital mortality, irrespective of other factors [91]. Elevated direct bilirubin has also been shown to independently predict higher risk of adverse events in patients with acute HF [92]. Similarly, clinical trials including the EVEREST, ASCEND-AHF, and STRONG-HF have confirmed the prognostic role of abnormal liver function tests in predicting adverse outcomes, such as rehospitalization and death [93,94,95]. Notably, congestion-related biomarkers (e.g., bilirubin, GGT, ALP) tend to normalise with decongestion, while biomarkers of hypoperfusion (e.g., low albumin, elevated AST/ALT) may indicate more severe hepatic injury [90]. Presently elevated liver biomarkers predict impaired long-term outcomes following hospitalisation for HF [96].

Chronic RV dysfunction may result in persistent hepatic congestion, progressive fibrosis, and ultimately cardiac cirrhosis, that can limit eligibility for advanced HF treatments such as transplantation or mechanical circulatory support [90]. Therefore, liver biomarkers play a crucial role in both the diagnostic assessment and prognostic evaluation of RV dysfunction.

### 4.7. Emerging Biomarkers

The search for novel biomarkers in HF is progressing rapidly, with growing emphasis on identifying those that more specifically reflect the underlying mechanisms of RV involvement. Among those under investigation, fibromodulin and fibulin-5 have shown potential specificity for RV dysfunction in HFrEF [97]. Fibroblast growth factor-23 (FGF-23) has demonstrated a strong association with RV dysfunction in both HFrEF and pulmonary hypertension, independently of congestion or BNP levels, and was associated with a risk of adverse outcomes [98,99]. SPARC-like protein 1 has been linked to maladaptive RV remodelling in pulmonary hypertension [100]. However, the clinical value of these and other biomarkers, such as dipeptidyl peptidase-3 (DPP3, a biomarker of inflammatory and oxidative stress associated with decreased myocardial contractility) in cardiogenic shock, requires validation in larger studies.

## 5. The Role of Biomarkers in the Management of Right Ventricular Dysfunction

In the setting of acute RV dysfunction, management is guided by three key principles: addressing the underlying cause (e.g., reperfusion in ST elevation myocardial infarction or fibrinolysis in pulmonary embolism), optimizing volume status and RV preload, and providing medical and/or mechanical circulatory support when needed. In chronic HF (HFrEF or HFpEF), available evidence supports possible improvement of RV function with the use of contemporary medications [101]. Biomarkers are increasingly being used in guiding clinical decisions, from monitoring therapeutic response in acute settings to facilitating risk stratification, transition of care, and follow-up of patients. Although their specificity for RV dysfunction remains limited, certain evidence supports their utility.

In the acute setting, patients with high-risk pulmonary embolism should receive systemic thrombolytic therapy, or surgical or percutaneous catheter-directed treatment if thrombolysis is contraindicated [36]. Administration of systemic thrombolysis is also recommended in patients with intermediate or intermediate-high risk features, based on clinical, imaging and biomarker assessment, who deteriorate on anticoagulant therapy [36].

High-sensitivity C-reactive protein was used in the CORTAHF trial to assess the therapeutic response to burst steroid therapy versus placebo in patients with acute HF and elevated inflammatory biomarkers [102]. Steroid treatment led to a marked reduction in inflammation, accompanied by functional improvement and a lower risk of adverse clinical outcomes [102]. In patients hospitalised for HF, a thorough evaluation of congestion is recommended, both at admission and prior to discharge, by incorporating clinical congestion scores, imaging techniques, and serial biomarker measurements [103]. Natriuretic peptides have been most extensively evaluated in guiding predischarge assessment of decongestion. A decrease in NT-proBNP levels of >30% from admission to discharge or achieving a predischarge NT-proBNP concentration < 1500 pg/mL, has been linked with lower risk of adverse events following discharge [104]. However, given the low specificity of natriuretic peptides for RV dysfunction, a possibility of using other biomarkers has gained interest. A multi-biomarker assessment has been proposed, where further evaluation with CA125, copeptin, h-FABP, and bio-ADM could complement predischarge natriuretic peptide assessment [6]. In addition, late and sustained increase in haematocrit before discharge along with evidence of clinical stabilisation, may indicate equilibration between intra- and extravascular fluid compartments and achievement of euvolemia [105].

In patients with severe RV dysfunction and cardiogenic shock, monitoring of renal and hepatic function and electrolyte levels is highly relevant to optimise the treatment. Furthermore, combined assessment of creatinine-based eGFR and NGAL, has been shown to provide earlier and more reliable detection of acute kidney injury in this setting [87]. Lactate monitoring is also recommended in patients with cardiogenic shock to assess treatment response, as persistently elevated levels are linked to poor prognosis [106]. Certain biomarkers are also being explored as novel therapeutic targets, including monoclonal antibodies against DPP3 (procizumab) and bio-ADM (adrecizumab) [107]. Although, a clinical trial assessing the efficacy of adrecizumab to reduce the need for cardiovascular support or improve short-term and intermediate survival showed neutral results [108], ongoing research is advancing the application of biomarkers and providing targeted management of cardiogenic shock [109].

While the effectiveness of natriuretic peptide-guided therapy following HF discharge remains debated [110], the CHANCE-HF trial yielded encouraging results for CA125-guided management. In this study, patients randomly assigned to receiving intensified follow-up and diuretic dose adjustments aimed at maintaining CA125 levels ≤ 35 U/mL were compared with those receiving standard care [111]. CA125-guided management significantly reduced the incidence of HF hospitalization or mortality at one year, primarily due to a reduction in HF hospitalisation [111]. More recently, findings from the STRONG-HF trial suggest that a strategy of GDMT up-titration early after discharge, utilising increased NT-proBNP as guidance to increase diuretic therapy and reduce the GDMT up-titration rate, proved successful for achieving clinical improvement [112]. Despite these promising results, further efforts are needed to provide specific therapeutic guidance for patients with RV dysfunction.

In advanced HF, biomarkers play a critical role in identifying and monitoring RV dysfunction, especially when invasive hemodynamic data are unavailable. Despite limited specificity of natriuretic peptide assessment, their combined assessment with other biomarkers, including those of liver (bilirubin, GGT, AST/ALT) and renal dysfunction (creatinine, eGFR, cystatin C) can signal organ congestion secondary to RV impairment. In patients with LVAD, pre-implant elevation of these biomarkers correlates with increased risk of early RV failure [113]. Post-LVAD, persistently abnormal biomarkers can suggest evolving RV dysfunction before clinical decompensation becomes evident. When used in conjunction with right heart catheterization data (i.e., central venous pressure, pulmonary artery pulsatility index, and right atrial pressure), biomarkers can enhance diagnostic precision and risk stratification [114]. This synergy is essential for early detection, management, and prognostication of RV dysfunction in both advanced HF and post-LVAD settings.

## 6. Conclusions and Future Directions

RV dysfunction is an underrecognized yet important contributor to adverse outcomes in a range of cardiovascular and systemic diseases. Although no biomarker has yet been identified as specific for RV dysfunction, several biomarkers gain relevance in defined clinical settings, where they can enhance diagnostic precision and risk stratification. Table 3 summarizes RV specificity, diagnostic, prognostic, and therapeutic relevance of the established biomarkers in the assessment of RV dysfunction.

While cardiac troponins and natriuretic peptides lack RV specificity, they are readily available and broadly used in clinical management. Novel and emerging biomarkers may improve diagnostic precision, but their optimal use remains unclear due to issues such as timing of measurement, confounding by age, renal function, and comorbidities, and uncertain correlation with disease progression. Clinical utility will require validation in randomized trials to demonstrate improved outcomes. The utility of novel and emerging biomarkers may be hampered by low availability in routine practice, significant costs, and lack of standardisation. Future efforts should focus on a validated, multi-biomarker strategy tailored to RV dysfunction [6]. AI-assisted decision support systems (AI-DSS) that integrate clinical, imaging, and biomarker data may further strengthen a multi-biomarker–tailored approach. By applying machine learning algorithms, these tools can perform cluster analyses of multiple biomarkers to define profiles with higher specificity for RV dysfunction [115]. When combined with clinical, electrocardiographic, and imaging data (e.g., echocardiography, CMR), AI-DSS could substantially improve early detection, diagnostic precision, and management of patients with RV dysfunction. Nonetheless, the development, validation, and clinical implementation of such systems remain areas for future research.

## Figures and Tables

**Figure 1 ijms-26-09064-f001:**
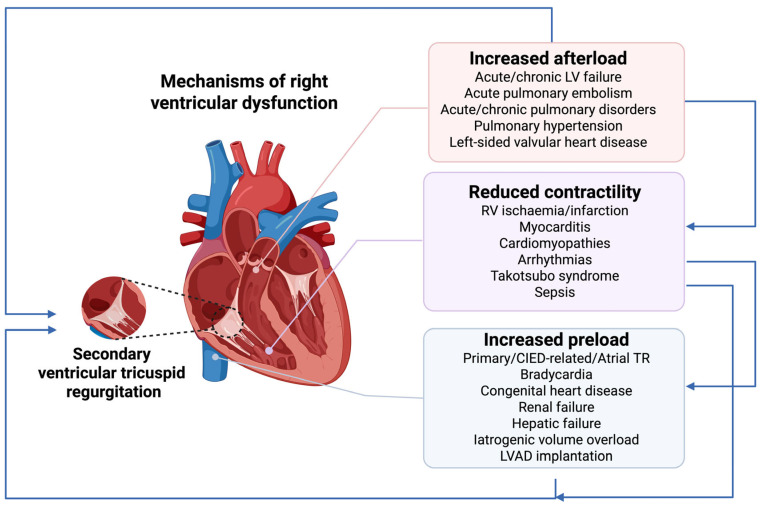
Aetiologias and mechanisms of right ventricular dysfunction. CIED—cardiac implantable devices, LV—left ventricular, LAVD—left ventricular assist device, RV—right ventricular, TR—tricuspid regurgitation. Note: RV dysfunction results from the interplay of increased afterload, reduced contractility, and excessive preload. Acute or chronic pressure overload leads to dilation, hypertrophy, and secondary ventricular tricuspid regurgitation. Reduced contractility leads to RV dilatation and secondary ventricular tricuspid regurgitation, which increased RV preload and exacerbates RV dysfunction. Excessive preload further strains the RV. Secondary tricuspid regurgitation amplifies all mechanisms, fueling congestion and creating a vicious cycle of RV failure. Interconnecting arrows depict how these mechanisms interact and collectively contribute to RV dysfunction progression.

**Figure 2 ijms-26-09064-f002:**
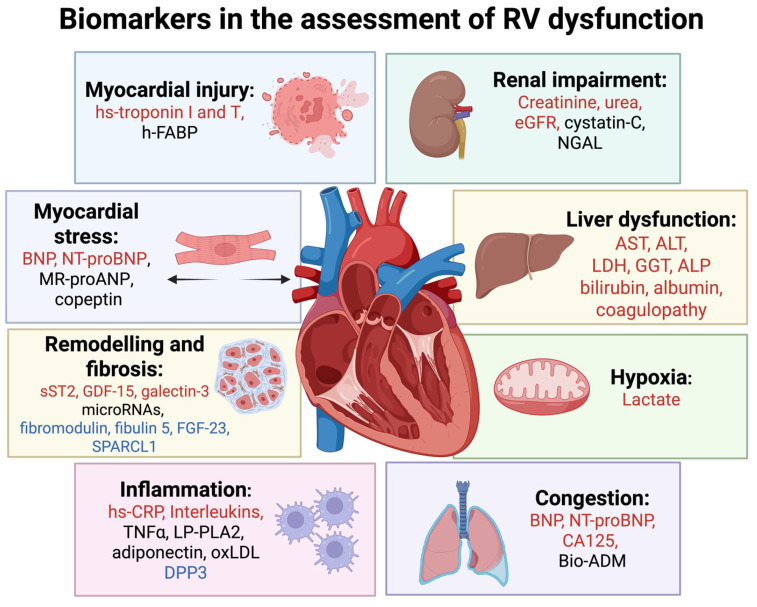
Established and emerging biomarkers reflecting pathophysiological processes relevant for RV dysfunction. ALT—alanine aminotransferase, ALP—alkaline phosphatase, AST—aspartate aminotransferase, Bio-ADM—biologically active adrenomedullin, BNP—B-type natriuretic peptide, CA-125—carbohydrate antigen 125, DPP3—dipeptidyl peptidase-3, eGFR—estimated glomerular filtration rate, FGF23—fibroblast growth factor 23, GDF-15—growth differentiation factor-15, GGT—gamma-glutamyl transferase hs-CRP—high sensitivity C-reactive protein, LDH—lactate dehydrogenase, LP-PLA2—lipoprotein-associated phospholipase A2, micro-RNAs—micro ribonucleic acids, MR-proANP—midregional pro-atrial natriuretic peptide, NGAL—neutrophil gelatinase-associated lipocalin, NT-proBNP—N-terminal pro-B type natriuretic peptide, oxLDL—oxidised low density lipoprotein, TNFα—tumour necrosis factor alfa, SPARCL1—SPARC-like protein 1, sST2—soluble suppression of tumorigenicity-2. Note: Established biomarkers with the strongest validation for diagnosis, prognosis, or therapy are shown in red; those with established but less robust validation in black; and emerging biomarkers in blue.

**Table 1 ijms-26-09064-t001:** Clinical and electrocardiographic signs of RV dysfunction **.

Clinical Symptoms and Signs	ECG Findings
Symptoms: ▪ Fatigue ▪ Exercise intolerance ▪ Palpitations ▪ Syncope	P-pulmonale
Systemic venous congestion: ▪ Distended jugular veins ▪ Pleural and pericardial effusions ▪ Hepato-renal congestion ▪ Ascites ▪ Peripheral oedema ▪ Anasarca	▪ S1Q3T3 (McGinn-White sign) ▪ Right axis deviation
Hypotension and hypoperfusion	Cardiac rhythm and conduction abnormalities: ▪ RBBB ▪ AV block ▪ Atrial fibrillation
Cyanosis	Low voltage QRS in the limb leads (pericardial/pleural effusion, amyloidosis)
Tachycardia	ST elevation and/or negative T wave in the precordial leads
Kussmaul’s sign (increased jugular venous pressure on inspiration)	Tall R waves in V1 and V2
Systolic heart murmur over tricuspid valve	Epsilon wave (ARVC)

AV—atrioventricular, ARVC—arrhythmogenic right ventricular cardiomyopathy, RBBB—right bundle branch block; ** according to references [5,12,20,22,23].

**Table 2 ijms-26-09064-t002:** Echocardiographic findings indicative of RV dysfunction.

Echocardiographic Findings
RVEDD/LVEDD > 1
RV basal diameter > 41 mm, mid-ventricular diameter > 35 mm, longitudinal diameter < 86 mm (measured from apical 4-chamber view)
RV free wall thickness > 5 mm
TAPSE < 17 mm
Systolic S′ velocity of the tricuspid valve annulus < 9.5 cm/s
RV fractional area change < 35%
RV index of myocardial performance < 0.54
Increased estimated RVSP
Significant TR
TR-V > 2.8 m/s
Abnormal IVC diameter and collapsibility (>21 mm diameter, <50% inspiratory collapsibility), VExUS
RVEF < 45% *
Impaired RV longitudinal strain by 2D speckle-tracking

LVEDD—left ventricle end-diastolic diameter; RV—right ventricle; EVEF—right ventricular ejection fraction; RVEDD—right ventricle end-diastolic diameter; TAPSE—tricuspid annular plane systolic excursion; RVSP—right ventricle systolic pressure; TR—tricuspid regurgitation; TR-V—tricuspid regurgitation jet velocity; VE × US—venous excess ultrasound; VCI—vena cava inferior. * Abnormal IVC diameter and collapsibility and abnormal VE × US score reflect venous congestion frequently associated with RV dysfunction. * RVEF should be assessed by three-dimensional echocardiography. According to references [5,12,20,22,23].

**Table 3 ijms-26-09064-t003:** A summary of the specificity and clinical utility of biomarkers in the assessment of RV dysfunction.

Biomarkers	Specificity for RV Dysfunction	Diagnostic Utility	Prognostic Utility	Utility in Treatment Guidance
**1. Myocardial injury**				
Cardiac troponins	Low	+++	+++	+++
hFABP	Moderate	+++	+++	To be defined
**2. Myocardial stress**				
B-type natriuretic peptides (BNP and NT-proBNP)	Low	+++	+++	+++
MR-proANP	Low	+	+	To be defined
Copeptin	Modest	++	++	To be defined
**3. Remodelling and fibrosis**				
sST2	Modest	+	+	To be defined
GDF-15	Low	+	+	To be defined
Galectin-3	Low	+	+	To be defined
Micro-RNAs	Low	+	+	To be defined
**4. Inflammation**				
hsCRP	Low	+++	+++	+++
Interleukins	Low	+++	+++	+++
TNFα	Low	+	+	+
PL-PLA2	Low	+	+	To be defined
Adiponectin	Low	+	+	To be defined
oxLDL	Low	+	+	To be defined
**5. Congestion**				
CA125	High (RV dysfunction > LV dysfunction)	+++	+++	+++
Bio-ADM	Modest	++	++	To be defined
**6. Hypoxia**				
Lactate	Low	+++	+++	To be defined
**7. Liver dysfunction**				
Hepatocellular damage/cholestasis (AST, ALT, ALP, GGT, bilirubin)	Modest	+++	+++	To be defined
Synthetic function (albumin, prothrombin time)	Modest	+++	+++	+++
**8. Renal impairment**				
Blood urea nitrogen, serum creatinine, eGFR	Low	+++	+++	+++
Cystatin-C	Modest (better than natriuretic peptides)	+++	+++	++
NGAL	Modest	++	++	To be defined

ALT—alanine aminotransferase, ALP—alkaline phosphatase, AST—aspartate aminotransferase, Bio-ADM—biologically active adrenomedullin, BNP—B-type natriuretic peptide, CA-125—carbohydrate antigen 125, eGFR—estimated glomerular filtration rate, GDF-15—growth differentiation factor-15, GGT—gamma-glutamyl transferase hs-CRP—high sensitivity C-reactive protein, LDH—lactate dehydrogenase, LP-PLA2—lipoprotein-associated phospholipase A2, micro-RNAs—micro ribonucleic acids, MR-proANP—midregional pro-atrial natriuretic peptide, NGAL—neutrophil gelatinase-associated lipocalin, NT-proBNP—N-terminal pro-B type natriuretic peptide, oxLDL—oxidised low density lipoprotein, TNFα—tumour necrosis factor alfa, sST2—soluble suppression of tumorigenicity-2. Note: The RV specificity, diagnostic and prognostic utility, and potential role in treatment guidance reflect the current evidence and studies discussed in this manuscript. Further research is required to better define and strengthen the clinical role of some of the presented biomarkers. Due to a paucity of supporting data, emerging biomarkers are not presented. Symbols denote strength of available evidence: + modest, ++ moderate, +++ strong.

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
