# Peer review of "The Right Approach: Power of Biomarkers in the Assessment and Management of Right Ventricular Dysfunction"

_ijms, 2025, doi:10.3390/ijms26189064_

Round 1
Reviewer 1 Report
Comments and Suggestions for Authors
This review provides a comprehensive and clinically relevant overview of biomarkers associated with right ventricular (RV) dysfunction in heart failure (HF). The manuscript is well-structured and informative. However, there are several concerns regarding the figures, particularly Figures 1 and 2, which require attention to improve clarity, consistency, and alignment with the text.
Figure 1 – Structural and Interpretive Issues Inconsistency with Textual Flow: The order of pathophysiological mechanisms in Figure 1 does not match the sequence presented in the manuscript. The text first discusses reduced contractility, followed by increased preload and increased afterload. However, the figure presents these elements in a different order, which may confuse readers trying to follow the narrative. Unexplained Elements: Figure 1 includes conditions such as Sepsis and Takotsubo syndrome under “reduced contractility,” which are not discussed in the corresponding section of the text. Their inclusion without explanation disrupts the coherence between the figure and the manuscript and should either be justified in the text or removed from the figure. Ambiguous Arrows and Flow: The arrows in Figure 1 lack clear meaning—whether they indicate causality, progression, or classification is unclear. Adding a legend or explanatory notes would help clarify the relationships depicted.
Figure 2 – Scope and Content Concerns Mismatch Between Figure Legend and Content:
The legend of Figure 2 states: “Established and emerging biomarkers reflecting pathophysiological process relevant for RV dysfunction.” However, emerging biomarkers discussed in Section 4.7—such as fibromodulin, fibulin-5, FGF-23, SPARC-like protein 1, and DPP3—are not represented in the figure. This creates a disconnect between the figure’s stated scope and its actual content.
Lack of Visual Hierarchy: All biomarkers are presented with equal visual weight, regardless of their clinical relevance or level of validation. Consider using color coding, grouping, or sizing to distinguish between established and emerging biomarkers, or between diagnostic and prognostic utility.
Minor Correction – Typographical Error
In line 65, the biomarker is incorrectly referred to as “galactin-3”.
The correct spelling is “galectin-3”.
Author Response
This review provides a comprehensive and clinically relevant overview of biomarkers associated with right ventricular (RV) dysfunction in heart failure (HF). The manuscript is well-structured and informative. However, there are several concerns regarding the figures, particularly Figures 1 and 2, which require attention to improve clarity, consistency, and alignment with the text.
Comment: Figure 1 – Structural and Interpretive Issues Inconsistency with Textual Flow: The order of pathophysiological mechanisms in Figure 1 does not match the sequence presented in the manuscript. The text first discusses reduced contractility, followed by increased preload and increased afterload. However, the figure presents these elements in a different order, which may confuse readers trying to follow the narrative. Unexplained Elements: Figure 1 includes conditions such as Sepsis and Takotsubo syndrome under “reduced contractility,” which are not discussed in the corresponding section of the text. Their inclusion without explanation disrupts the coherence between the figure and the manuscript and should either be justified in the text or removed from the figure. Ambiguous Arrows and Flow: The arrows in Figure 1 lack clear meaning—whether they indicate causality, progression, or classification is unclear. Adding a legend or explanatory notes would help clarify the relationships depicted.
Response:
<<< We thank the Reviewer for the important comments and practical suggestions which we have implemented to improve the quality, consistency and flow of the manuscript.
We have reordered the text so that the sequence of pathophysiological processes matches those presented in Figure 1. In addition, all mechanisms illustrated in Figure 1 are addressed in the accompanying text, including sepsis and takotsubo syndrome in the section on reduced contractility:
“1. Increased afterload
The RV is sensitive to an increase in afterload due to its relationship with the pulmonary vascular network, which is normally characterised by low vascular resistance. In the acute setting (i.e. acute HF, pulmonary embolism, pneumonia, acute respiratory distress), the RV responds by dilating and increasing its contractility, but RV failure may occur rapidly if these mechanisms prove insufficient (5). In the chronic setting (i.e. chronic left-sided HF, pulmonary hypertension, chronic pulmonary disorders), the RV undergoes progressive hypertrophy, enabling it to preserve cardiac output. However, it eventually dilates, resulting in tricuspid regurgitation and a subsequent decline in cardiac output (5). This adaptive response is particularly relevant in chronic states associated with pulmonary hypertension, where the RV-pulmonary artery coupling is essential for maintaining cardiac output (13).
- Reduced contractility
Acute reduction in RV contractility is mostly caused by acute myocardial ischemia/infarction or acute inflammation (i.e. myocarditis) (14, 15). In sepsis and septic shock, the systemic inflammatory response was associated with a high prevalence (48%) of RV dysfunction, defined by echocardiographically assessed fractional area change <35% or tricuspid annular plane systolic excursion <16 mm (16). RV dysfunction is associated with a more complicated clinical course and reduced 30-day survival (16, 17). Moreover, RV involvement has been reported in patients with left-sided takotsubo syndrome, with isolated RV forms also documented, typically presenting with more severe clinical manifestations (18, 19). Chronic RV dysfunction most commonly accompanies left-sided HF but may also result from chronic pressure or volume overload or intrinsic myocardial diseases such as cardiomyopathies or cardiac sarcoidosis (20). In addition to reduced cardiac output, dilated and hypocontractile RV contributes to left ventricular (LV) dysfunction through ventricular interdependence. The overloaded RV "competes" with the LV for the space within the pericardium, leading to the displacement of the interventricular septum towards the LV, impairment in LV filling, and further reduction in cardiac output. Furthermore, elevated pressures in RV free wall, and a decrease in systemic blood pressure, can impact the coronary blood flow, and precipitate worsening RV failure (12).
- Increased preload
The RV is highly dependent on the appropriate preload for its optimal function. However, excessive RV preload due to renal and/or hepatic failure, severe tricuspid regurgitation, congenital heart diseases (i.e. atrial or ventricular septal defect, partial or total anomalous pulmonary venous return, Ebstein anomaly, patent ductus arteriosus) or iatrogenic volume overload can lead to deterioration in RV function (12). Significant bradycardia can also contribute to an increase in RV preload. In addition, increased LV output and venous return following implantation of a left ventricular assist device acutely increase RV preload and can precipitate RV failure.”
A note has been included for Figure 1 to explain the interconnection of the pathophysiological processes driving RV dysfunction and the role of the arrows linking those processes:
“Note: RV dysfunction results from the interplay of increased afterload, reduced contractility, and excessive preload. Acute or chronic pressure overload leads to dilation, hypertrophy, and secondary ventricular tricuspid regurgitation. Reduced contractility leads to RV dilatation and secondary ventricular tricuspid regurgitation, which increased RV preload and exacerbates RV dysfunction. Excessive preload further strains the RV. Secondary tricuspid regurgitation amplifies all mechanisms, fuelling congestion and creating a vicious cycle of RV failure. Interconnecting arrows depict how these mechanisms interact and collectively contribute to RV dysfunction progression”.>>>
Figure 2 – Scope and Content Concerns Mismatch Between Figure Legend and Content:
Comment: The legend of Figure 2 states: “Established and emerging biomarkers reflecting pathophysiological process relevant for RV dysfunction.” However, emerging biomarkers discussed in Section 4.7—such as fibromodulin, fibulin-5, FGF-23, SPARC-like protein 1, and DPP3—are not represented in the figure. This creates a disconnect between the figure’s stated scope and its actual content.
Response:
<<<We thank the Reviewer for noticing the discrepancy. We added emerging biomarkers with appropriate colour coding in Figure 2 (kindly see our response to the comment below).>>>
Comment: Lack of Visual Hierarchy: All biomarkers are presented with equal visual weight, regardless of their clinical relevance or level of validation. Consider using color coding, grouping, or sizing to distinguish between established and emerging biomarkers, or between diagnostic and prognostic utility.
Response:
<<< We agree with the Reviewer on this important comment. We have included emerging biomarkers in Figure 2. and revised the figure in hierarchical order to first present established and best validated biomarkers (color coded in red), then established but less validated biomarkers (black) and lastly emerging biomarkers (blue). A note is added below the Figure explaining the color coding:
“Note: Established biomarkers with the strongest validation for diagnosis, prognosis, or therapy are shown in red; those with established but less robust validation in black; and emerging biomarkers in blue“
We could not differentiate and provide specific coding for diagnostic vs prognostic role, since most biomarkers have both, and several have also shown value in guiding therapy, which is further discussed in the text. However, this issue is further addressed in Table 3 (provided in the Conclusions and Future Directions):
Table 3. A summary of the specificity and clinical utility of biomarkers in the assessment of RV dysfunction
|
Biomarkers |
Specificity for RV dysfunction |
Diagnostic utility |
Prognostic utility |
Utility in treatment guidance |
|
1. Myocardial injury |
|
|
|
|
|
Cardiac troponins |
Low |
+++ |
+++ |
+++ |
|
hFABP |
Moderate |
+++ |
+++ |
To be defined |
|
2. Myocardial stress |
|
|
|
|
|
B-type natriuretic peptides (BNP and NT-proBNP) |
Low |
+++ |
+++ |
+++ |
|
MR-proANP |
Low |
+ |
+ |
To be defined |
|
Copeptin |
Modest |
++ |
++ |
To be defined |
|
3. Remodelling and fibrosis |
|
|
|
|
|
sST2 |
Modest |
+ |
+ |
To be defined |
|
GDF-15 |
Low |
+ |
+ |
To be defined |
|
Galectin-3 |
Low |
+ |
+ |
To be defined |
|
Micro-RNAs |
Low |
+ |
+ |
To be defined |
|
4. Inflammation |
|
|
|
|
|
hsCRP |
Low |
+++ |
+++ |
+++ |
|
Interleukins |
Low |
+++ |
+++ |
+++ |
|
TNFα |
Low |
+ |
+ |
+ |
|
PL-PLA2 |
Low |
+ |
+ |
To be defined |
|
Adiponectin |
Low |
+ |
+ |
To be defined |
|
oxLDL |
Low |
+ |
+ |
To be defined |
|
5. Congestion |
|
|
|
|
|
CA125 |
High (RV dysfunction > LV dysfunction) |
+++ |
+++ |
+++ |
|
Bio-ADM |
Modest |
++ |
++ |
To be defined |
|
6. Hypoxia |
|
|
|
|
|
Lactate |
Low |
+++ |
+++ |
To be defined |
|
7. Liver dysfunction |
|
|
|
|
|
Hepatocellular damage/cholestasis (AST, ALT, ALP, GGT, bilirubin) |
Modest |
+++ |
+++ |
To be defined |
|
Synthetic function (albumin, prothrombin time) |
Modest |
+++ |
+++ |
+++ |
|
8. Renal impairment |
|
|
|
|
|
Blood urea nitrogen, serum creatinine, eGFR |
Low |
+++ |
+++ |
+++ |
|
Cystatin-C |
Modest (better than natriuretic peptides) |
+++ |
+++ |
++ |
|
NGAL |
Modest |
++ |
++ |
To be defined |
ALT – alanine aminotransferase, ALP – alkaline phosphatase, AST – aspartate aminotransferase, Bio-ADM – biologically active adrenomedullin, BNP – B-type natriuretic peptide, CA-125 – carbohydrate antigen 125, eGFR – estimated glomerular filtration rate, GDF-15 – growth differentiation factor-15, GGT - gamma-glutamyl transferase hs-CRP – high sensitivity C-reactive protein, LDH - lactate dehydrogenase, LP-PLA2 - lipoprotein-associated phospholipase A2, micro-RNAs – micro ribonucleic acids, MR-proANP - midregional pro-atrial natriuretic peptide, NGAL - neutrophil gelatinase-associated lipocalin, NT-proBNP – N-terminal pro-B type natriuretic peptide, oxLDL – oxidised low density lipoprotein, TNFα – tumour necrosis factor alfa, sST2 – soluble suppression of tumorigenicity-2.
Note: The RV specificity, diagnostic and prognostic utility, and potential role in treatment guidance reflect the current evidence and studies discussed in this manuscript. Further research is required to better define and strengthen the clinical role of some of the presented biomarkers. Due to a paucity of supporting data, emerging biomarkers are not presented. Symbols denote strength of available evidence: + modest, ++ moderate, +++ strong.>>>
Minor Correction – Typographical Error
Comment: In line 65, the biomarker is incorrectly referred to as “galactin-3”.
The correct spelling is “galectin-3”.
Response:
<<<We thank the Reviewer. The typo has been corrected.>>>
We kindly ask the Reviewer to take a look at the enclosed word document with our responses, which also includes the Revised Figure 2.

Reviewer 2 Report
Comments and Suggestions for Authors
This review, “The Right Approach: Power of Biomarkers in the Assessment and Management of Right Ventricular Dysfunction,” is well-written, timely, and covers an important area that often receives less attention than left-sided heart disease. The manuscript is comprehensive, with a clear structure and a strong reference base, and it succeeds in bringing together epidemiology, pathophysiology, diagnostic tools, and therapeutic perspectives. The tables and figures are helpful for readers navigating a large body of information.
-
I suggest to strengthen the discussion on biomarker specificity by comparing RV-targeted biomarkers with systemic ones, to better show differences in diagnostic value.
-
I suggest to highlight the practical limitations of emerging biomarkers (e.g., cost, availability, lack of standardization), so readers can better understand their current clinical applicability.
-
I suggest to reorganize Figure 2, grouping biomarkers by clinical utility (diagnosis, prognosis, therapy) rather than only by pathophysiological processes.
-
I suggest to expand the future outlook, with more concrete directions on multi-biomarker strategies, integration with imaging, and the role of AI.
-
I suggest to improve style and readability by shortening some sentences and polishing phrasing for smoother flow.
Overall, this is an excellent review that pulls together a wide range of information in a field that is clinically important yet often underexplored. With some minor adjustments, it will make a very valuable contribution.
Author Response
This review, “The Right Approach: Power of Biomarkers in the Assessment and Management of Right Ventricular Dysfunction,” is well-written, timely, and covers an important area that often receives less attention than left-sided heart disease. The manuscript is comprehensive, with a clear structure and a strong reference base, and it succeeds in bringing together epidemiology, pathophysiology, diagnostic tools, and therapeutic perspectives. The tables and figures are helpful for readers navigating a large body of information.
Comment: I suggest to strengthen the discussion on biomarker specificity by comparing RV-targeted biomarkers with systemic ones, to better show differences in diagnostic value.
Response:
<<< We thank the Reviewer for this valuable comment. In response, we have added Table 3 in the Conclusions and Future Directions to summarise the RV specificity and clinical utility of the biomarkers discussed, including their diagnostic, prognostic, and therapeutic roles
“Although no biomarker has yet been identified as specific for RV dysfunction, several biomarkers gain relevance in defined clinical settings, where they can enhance diagnostic precision and risk stratification. Table 3 summarizes RV specificity, diagnostic, prognostic, and therapeutic relevance of the established biomarkers in the assessment of RV dysfunction.
Table 3. A summary of the specificity and clinical utility of biomarkers in the assessment of RV dysfunction
|
Biomarkers |
Specificity for RV dysfunction |
Diagnostic utility |
Prognostic utility |
Utility in treatment guidance |
|
1. Myocardial injury |
|
|
|
|
|
Cardiac troponins |
Low |
+++ |
+++ |
+++ |
|
hFABP |
Moderate |
+++ |
+++ |
To be defined |
|
2. Myocardial stress |
|
|
|
|
|
B-type natriuretic peptides (BNP and NT-proBNP) |
Low |
+++ |
+++ |
+++ |
|
MR-proANP |
Low |
+ |
+ |
To be defined |
|
Copeptin |
Modest |
++ |
++ |
To be defined |
|
3. Remodelling and fibrosis |
|
|
|
|
|
sST2 |
Modest |
+ |
+ |
To be defined |
|
GDF-15 |
Low |
+ |
+ |
To be defined |
|
Galectin-3 |
Low |
+ |
+ |
To be defined |
|
Micro-RNAs |
Low |
+ |
+ |
To be defined |
|
4. Inflammation |
|
|
|
|
|
hsCRP |
Low |
+++ |
+++ |
+++ |
|
Interleukins |
Low |
+++ |
+++ |
+++ |
|
TNFα |
Low |
+ |
+ |
+ |
|
PL-PLA2 |
Low |
+ |
+ |
To be defined |
|
Adiponectin |
Low |
+ |
+ |
To be defined |
|
oxLDL |
Low |
+ |
+ |
To be defined |
|
5. Congestion |
|
|
|
|
|
CA125 |
High (RV dysfunction > LV dysfunction) |
+++ |
+++ |
+++ |
|
Bio-ADM |
Modest |
++ |
++ |
To be defined |
|
6. Hypoxia |
|
|
|
|
|
Lactate |
Low |
+++ |
+++ |
To be defined |
|
7. Liver dysfunction |
|
|
|
|
|
Hepatocellular damage/cholestasis (AST, ALT, ALP, GGT, bilirubin) |
Modest |
+++ |
+++ |
To be defined |
|
Synthetic function (albumin, prothrombin time) |
Modest |
+++ |
+++ |
+++ |
|
8. Renal impairment |
|
|
|
|
|
Blood urea nitrogen, serum creatinine, eGFR |
Low |
+++ |
+++ |
+++ |
|
Cystatin-C |
Modest (better than natriuretic peptides) |
+++ |
+++ |
++ |
|
NGAL |
Modest |
++ |
++ |
To be defined |
ALT – alanine aminotransferase, ALP – alkaline phosphatase, AST – aspartate aminotransferase, Bio-ADM – biologically active adrenomedullin, BNP – B-type natriuretic peptide, CA-125 – carbohydrate antigen 125, eGFR – estimated glomerular filtration rate, GDF-15 – growth differentiation factor-15, GGT - gamma-glutamyl transferase hs-CRP – high sensitivity C-reactive protein, LDH - lactate dehydrogenase, LP-PLA2 - lipoprotein-associated phospholipase A2, micro-RNAs – micro ribonucleic acids, MR-proANP - midregional pro-atrial natriuretic peptide, NGAL - neutrophil gelatinase-associated lipocalin, NT-proBNP – N-terminal pro-B type natriuretic peptide, oxLDL – oxidised low density lipoprotein, TNFα – tumour necrosis factor alfa, sST2 – soluble suppression of tumorigenicity-2.
Note: The RV specificity, diagnostic and prognostic utility, and potential role in treatment guidance reflect the current evidence and studies discussed in this manuscript. Further research is required to better define and strengthen the clinical role of some of the presented biomarkers. Due to a paucity of supporting data, emerging biomarkers are not presented. Symbols denote strength of available evidence: + modest, ++ moderate, +++ strong.>>>
Comment: I suggest to highlight the practical limitations of emerging biomarkers (e.g., cost, availability, lack of standardization), so readers can better understand their current clinical applicability.
Response:
<<<We agree with the comment. Accordingly, we made a comment about the practical limitation of novel and emerging biomarkers:
“Novel and emerging biomarkers may improve diagnostic precision, but their optimal use remains unclear due to issues such as timing of measurement, confounding by age, renal function, and comorbidities, and uncertain correlation with disease progression. Clinical utility will require validation in randomized trials to demonstrate improved outcomes. The utility of novel and emerging biomarkers may be hampered by low availability in routine practice, significant costs, and lack of standardisation.”>>>
Comment: I suggest to reorganize Figure 2, grouping biomarkers by clinical utility (diagnosis, prognosis, therapy) rather than only by pathophysiological processes.
Response:
<<<We thank the Reviewer on this important comment. However, in order to accommodate for the comment made by another Reviewer we have not regrouped Figure 2, but instead added Table 3. that summarises the clinical utility of biomarkers in RV assessment (kindly see Table 3, above). However, Figure 2 has been modified to improve distinction of the more established from emerging biomarkers.>>>
Comment: I suggest to expand the future outlook, with more concrete directions on multi-biomarker strategies, integration with imaging, and the role of AI.
Response:
<<<We are grateful for the important comment. In the Conclusions and Future Directions we added a paragraph on the integration of AI-based DSS in RV assessment:
“Future efforts should focus on a validated, multi-biomarker strategy tailored to RV dysfunction (115). AI-assisted decision support (AI-DSS) systems that integrate clinical, imaging, and biomarker data may further strengthen a multi-biomarker–tailored approach. By applying machine learning algorithms, these tools can perform cluster analyses of multiple biomarkers to define profiles with higher specificity for RV dysfunction (116). When combined with clinical, electrocardiographic, and imaging data (e.g., echocardiography, CMR), AI-DSS could substantially improve early detection, diagnostic precision, and management of patients with RV dysfunction. Nonetheless, the development, validation, and clinical implementation of such systems remain areas for future research.”>>>
Comment: I suggest to improve style and readability by shortening some sentences and polishing phrasing for smoother flow.
Response:
<<< We thank the Reviewer for this valuable comment. Where appropriate, we have revised and shortened sentences to enhance flow and readability.>>>
Comment: Overall, this is an excellent review that pulls together a wide range of information in a field that is clinically important yet often underexplored. With some minor adjustments, it will make a very valuable contribution.
Response:
<<< We thank the Reviewer for the encouraging words and valuable comments, which have helped us improve the manuscript>>>.
We kindly invite the Reviewer to consider the enclosed word document with our responses to the comments, which also includes the revised Figure 2.
